# Determining the Biological Features of Aggressive Meningioma Growth with Transcriptomic Profiling

**DOI:** 10.3390/cancers17203324

**Published:** 2025-10-15

**Authors:** Szymon Baluszek, Paulina Kober, Izabella Myśliwy, Artur Oziębło, Tomasz Mandat, Mateusz Piotr Jeżewski, Mateusz Bujko

**Affiliations:** 1Department of Molecular and Translational Oncology, Maria Sklodowska-Curie National Research Institute of Oncology, 02-781 Warsaw, Poland; szymon.baluszek@nio.gov.pl (S.B.); paulina.kober@nio.gov.pl (P.K.); izabella.mysliwy@nio.gov.pl (I.M.); 2Department of Neurosurgery, Maria Sklodowska-Curie National Research Institute of Oncology, 02-781 Warsaw, Poland; artur.ozieblo@nio.gov.pl (A.O.); tomasz.mandat@nio.gov.pl (T.M.); mateusz.jezewski@nio.gov.pl (M.P.J.)

**Keywords:** meningioma, tumor grade, genes expression, tumor microenvironment, border-associated macrophages

## Abstract

Meningiomas are frequently diagnosed tumors of intracranial location. In the WHO classification, tumor aggressiveness is assessed by assigning increasing grades, with WHO grade I tumors considered benign and WHO grade III tumors the most aggressive. In our study, we aimed to identify biological features associated with meningioma aggressiveness by profiling gene expression in tumor samples of different WHO grades (I, II, and III). We first identified genes with grade-related expression and subsequently analyzed their roles through biological pathway analysis and by estimating the cellular composition of the heterogeneous tumor samples. The results reflect well-established features of aggressive growth, such as increased cell proliferation, but also highlight less well-characterized aspects of tumor biology including alterations in cellular metabolism involving phosphoglycerate kinase 1, changes in transmembrane ion transport, and differences in the abundance of border-associated macrophages, which represent key distinctions between benign and aggressive meningiomas.

## 1. Introduction

Meningiomas are among the most common intracranial tumors in adults, accounting for approximately 38% of all tumors in this location [1]. They are diagnosed more frequently in people over the age of 65 and significantly more often in women, while remaining rare in pediatric patients.

Meningiomas encompass a broad histopathological spectrum, with approximately 80% meeting the diagnostic criteria for slow-growing, benign neoplasms classified as WHO (World Health Organization) grade I (GI) [2]. WHO grade II (GII) tumors, predominantly atypical meningiomas, and grade III (GIII) tumors, predominantly anaplastic meningiomas, account for approximately 18.3% and 1.6% of all meningiomas, respectively [1]. The WHO histopathological grade has long served as the principal determinant of prognosis and recurrence risk stratification factor. Patients with benign meningiomas generally experience a favorable clinical outcome, and surgical resection is effective in most cases [1]. Nonetheless, in some cases these benign tumors may progress to more aggressive forms, particularly following incomplete resection or radiation. WHO grade II and III meningiomas exhibit a more aggressive biological behavior, with a higher likelihood of recurrence, invasion of adjacent meninges, skull, or brain tissue, and are associated with reduced overall survival [3]. Histological criteria of aggressive meningiomas include increased mitotic activity and brain invasion as features of atypical meningiomas and high mitotic index (>20%), and nuclear atypia, necrosis and overtly malignant cytology being characteristics of anaplastic meningioma [2]. These histological features reflect underlying molecular changes, such as chromosomal instability, dysregulated cell cycle control, and epigenetic dysregulation.

In the recent years, scientific efforts have focused on characterizing molecular changes in meningiomas to elucidate potential pathogenic mechanisms and identifying markers of aggressive tumor growth and patients outcome [4,5,6]. Consequently, new molecular markers have been incorporated into the WHO classification to complement the histological criteria. Among these, *TERT* promoter mutations and homozygous deletions of *CDKN2A/B* play a particularly important role in classifying a meningioma as WHO grade III, even in the absence overtly anaplastic histology [2]. Comprehensive characterization of the transcriptomic and molecular landscape of meningiomas is crucial for uncovering the mechanisms underlying tumor aggressiveness and recurrence.

In this study, we used RNA sequencing (RNA-seq) to characterize transcriptomic difference among the main histological subtypes of WHO GI, GII, and GIII meningiomas. Our goal was to investigate differential genes expression and its functional relevance as well as to compare intratumoral heterogeneity between benign and high-grade meningiomas using deconvolution analysis.

## 2. Materials and Methods

### 2.1. Patients and Tissue Samples

A total of 60 formalin-fixed, paraffin-embedded (FFPE) meningioma specimens were obtained from patients who underwent surgical resection at the Maria Skłodowska-Curie National Research Institute of Oncology. The study group comprised 30 WHO grade I, 18 grade II, and 12 grade III tumors. All specimens underwent histopathological evaluation to confirm diagnosis of particular meningioma subtype and to select tumor subtype-representative sections for subsequent molecular analyses. Tissue samples were preselected to cover the most common histological subtypes of benign meningiomas: meningothelial (19) and fibrous (*n* = 11) as well as possibly large groups of the most common subtype of GII and GIII tumors. Transitional GI meningiomas were intentionally excluded since they are, by definition, characterized by mixed histological pattern. Less frequent histological subtypes were excluded to ensure relatively homogeneous study groups. The groups were designed to investigate differences between meningiomas of different grades, including two common and histologically distinct grade I subtypes. Patients’ characteristics are summarized in Table 1. RNA and DNA were extracted from FFPE material using the RecoverAll™ Total Nucleic Acid Isolation Kit for FFPE (Thermo Fisher Scientific, Waltham, MA, USA), and nucleic acid concentrations were determined with a NanoDrop 2000 spectrophotometer (Thermo Fisher Scientific). RNA was stored at −70 °C until analysis.

### 2.2. RNA Sequencing (RNA-Seq)

RNA isolated from 60 FFPE meningioma samples were used for RNA sequencing (RNA-seq). RNA quality was evaluated using the RNA 6000 Nano assay (Agilent Technologies, Santa Clara, CA, USA) run on the 2100 Bioanalyzer (Agilent Technologies). All the samples included in RNA-seq met the quality criterion of DV200 of at least 30%. Total RNA (100 ng) was used for library preparation using the SMART-Seq Stranded Kit (Takara Bio, San Jose, CA, USA) including ribosomal depletion step. Library quality was evaluated using the Agilent 2100 Bioanalyzer (Agilent Technologies, CA, USA). High-throughput sequencing was performed by CeGaT GmbH (Tübingen, Germany) using the Illumina (San Diego, CA, USA) NovaSeq 6000 platform, producing 100 bp paired-end reads with a minimum yield of 30 million read pairs per sample. Sequencing was performed by CeGaT GmbH.

### 2.3. Analysis of RNA Sequencing Data

Sequences retrieved from FFPE samples are known to undergo degradation, in particular on the 3′ end of the transcript [7]. Therefore, a custom approach was adapted (https://github.com/SBaluszek/miRinMeningioma (accessed on 5 January 2024)). First, reads were aligned to hg38.p11 genome with STAR software [8]. Next, quality control was performed with RseQC, and transcript integrity numbers (TINs) were obtained. Samples with less than 15% of reads aligned to genome were discarded and reads were normalized with DESeq2 [9]; TIN, fraction of reads aligned to genome and median reads per gene were treated as batch effects.

A normalized, scaled read count matrix was used for hierarchical clustering (Euclidian’s distance and ward.D agglomeration) and dimensionality reduction with uniform manifold approximation and projection (UMAP). Differentially expressed genes were obtained for tumor grade which were treated as numeric values for DESeq2 algorithm to follow. Gene Set Enrichment Analysis (GSEA) was conducted with fgsea [10]. Genes expression for the selected pathways was calculated and visualized using GSEAlm [11].

Finally, String DB [12], a protein–protein interaction (PPI) database was utilized to search for most important genes from a functional point of view; tidygraph [https://tidygraph.data-imaginist.com (accessed on 5 January 2024)] was utilized for visualization and estimating most central nodes in the PPI network.

### 2.4. Analysis of Publicly Available Single-Cell RNA-Sequencing Experiment

Count matrix for GSE183655 [13] was downloaded, loaded into Seurat (v5.1.0) [14] and subjected to quality control (number of features, number of counts, fraction of mitochondrial genes) and two samples (with patient ID MSC-6, one tumor, one brain–tumor interface) were rejected due to poor quality. Next, scds package [15] was utilized to identify doublets and cells with at least 1000 reads, of 500 unique features and with fraction of mitochondrial genes below 15% and doubled scds score below 1.5 were selected. Then, variable features were selected, utilizing cv2 algorithm [16] and samples were integrated utilizing the Harmony algorithm. Dimensional reduction was performed with uniform manifold approximation and projection (UMAP) (non-randomness of PCs was estimated, utilizing the Marchenko–Pastur algorithm [17]. Cell type clusters were identified, utilizing well-established markers. BisqueRNA [18] was utilized to estimate cell type population abundance in bulk RNA-seq samples. Subsequently, CellChat [19] was utilized to establish cell–cell interactions in the tumor, brain–tumor interface, and the dura. Weight of interactions and change in the weight of interactions were assessed. Cell-types and genes involved in this signaling were identified and visualized.

### 2.5. Immunohistochemistry (IHC)

Immunohistochemistry (IHC) was carried out on 4-µm FFPE tissue sections by the use of EnVision Detection System (DAKO, Glostrup, Denmark). Twenty-four tumor samples were subjected to immunohistochemical staining, with six samples representing each histological subtype: meningothelial, fibrous, atypical, and anaplastic meningiomas. Sections were deparaffinized in xylene and rehydrated through a graded ethanol series. Epitope retrieval was achieved by heating the slides in Target Retrieval Solution (pH 9, DAKO) at 96 °C for 30 min. Endogenous peroxidase activity was subsequently quenched with DAKO Peroxidase Blocking Reagent for 5 min. Slides were then incubated at room temperature for 1 h with mouse monoclonal anti-PGK1 antibody (clone 14, sc-130335, Santa Cruz Biotechnology (Dallas, TX, USA); 1:100 dilution). Detection was performed using 3,3′-diaminobenzidine tetrahydrochloride (DAB; DAKO) as chromogen, and nuclei were counterstained with hematoxylin. Stained tissue sections were examined at 200× magnification using Leica DEM 2000 LED microscope (Wetzlar, Germany). Immunoreactivity was assessed as a categorical variable using a four-point scale. Absence of expression was scored as 0, whereas low, moderate, and high expression were assigned scores of 1, 2, and 3, respectively. Analysis of proportion for statistical evaluation.

### 2.6. Statistical Analysis

For the analysis of continuous variables, the Shapiro–Wilk test was used to assess normality. Data with a Gaussian distribution were analyzed using one-way ANOVA, whereas the Kruskal–Wallis test was applied to non-normally distributed data. Two-sided Fisher’s exact test was applied for the analysis of categorical variables. Significance threshold of α = 0.05 was applied. Data was analyzed and visualized using GraphPad Prism v6.07 (GraphPad Software).

## 3. Results

### 3.1. Genes Expression Profiles in Benign and High-Grade Meningiomas

Four out of sixty samples were excluded from the analysis based on data quality control; therefore, fifty-six samples were included. Main parameter was fraction of reads aligned to the genome ([0.08–0.30], median: 0.20, IQR: 0.18–0.24). The of RNA-seq data were characterized by the median transcript integrity number (TIN) value of 67.59 (range 36.38–79.41, IQR: 59.54–72.97), and median count value of 12.00 (range 1.00–44.00, IQR: 8.00–21.00). Sequencing reads were mapped to 62,710 human transcripts. Total number of 42,596 transcripts met the filtering thresholds of a minimum of five reads in at least ten samples and were used for normalization and differential genes expression analysis.

Unsupervised analysis with dimensional reduction methods and hierarchical clustering revealed three clusters of samples that generally correspond to histological subtypes of meningiomas. Cluster M was mainly composed of meningothelial WHO GI meningiomas, Cluster F primarily of fibrous WHO GI tumors, and Cluster A mostly of aggressive WHO GII and GIII meningiomas, as depicted in Figure 1A,B. By differential analysis, using tumor WHO grading as numeric values, we identified 5518 protein-coding genes significantly associated with grade, as shown in Figure 1C and listed in Appendix A. The grade-related genes identified with this approach were prioritized as differentially expressed genes (DEGs) in the analysis of changes in gene expression levels between benign and anaplastic tumors.

Gene Set Enrichment Analysis (GSEA) with Gene Ontology (GO Biological Processes (GOBP) and Molecular Function (GOMF)) was applied for the functional interpretation of the identified DEGs. We found 1199 GOBP and 232 GOMF terms enriched for these genes (Appendix A). The processes most significantly enriched in higher-grade meningiomas included those related to cell proliferation, ribosome generation, and cellular metabolism, while those most significantly enriched in benign meningiomas were foremostly those of cell morphogenesis, transmembrane transport, and immune regulation as visualized in Figure 1D,E.

The most pronounced differences in processes related to cell mitosis reflect the use of proliferation assessment as a diagnostic criterion in meningioma grading. In contrast, we identified increased cellular metabolism and ribosome biogenesis, as well as decreased ion transport, as much less explored features of higher-grade meningiomas. Top DEGs, including *PGK1*, *DLAT*, *SHMT2*, *TSR1*, *C1QBP*, *BOP1*, *KCNMA1*, *CACNA1D*, *CACNA2D1*, *NUMB*, *WASF2* and *BVES*, were involved in these processes, as shown in Figure 2A. *PGK1*, a gene involved in metabolism-related processes, was the most significantly differentially expressed gene in meningiomas stratified according to the WHO tumor grade, as depicted in Figure 1C. Expression of phosphoglycerate kinase 1 (encoded by *PGK1*) was examined in set of a twenty-four meningioma samples, comprising six meningothelial, six fibrous, six atypical and six anaplastic meningiomas. PGK1 protein was expressed in each tumor sample; however, its clear grade-related increase in expression levels was observed with the highest expression observed in anaplastic meningiomas. Immunohistochemical staining (IHC) intensity was scored as weak (1), moderate (2), or strong (3) in tissue samples, and significant differences in the distribution of staining scores were identified among the benign (meningothelial and fibrous), atypical, and anaplastic meningioma groups (Fisher’s exact test, *p* = 0.0047). The majority of grade I tumors showed low expression, whereas grade III meningiomas displayed high PGK1 expression. IHC results are summarized in Table 2, while the representative examples of IHC staining images are presented Figure 2B.

### 3.2. Tumor Microenvironment and Cellular Communication in Benign and Aggressive Meningiomas

We used publicly available scRNA-seq data from meningioma samples (GSE183655) to determine the cellular composition of meningioma. Twenty-two cellular clusters were identified using well-established gene expression markers (Figure 3B). These clusters were assigned to known cellular components, including tumor cells, fibroblasts, endothelial cells, vascular cells, Schwann cells, border-associated macrophages (BAMs), monocytes, granulocytes, T-cells, B-cells, and microglial and glial cells (Figure 3A,B). We then used the transcriptomic profiles of these defined cellular components to deconvolute our RNA-seq data from meningioma samples of distinct histological subtypes (Figure 3C). Comparing tumor samples of benign, atypical and anaplastic meningiomas revealed grade-related differences in the microenvironment, particularly in the proportions of BAMs and granulocytes (Figure 3C). Notably, there was a gradual decrease in BAM content in tumors with increasing WHO grade. The median BAMs proportion was 23.12% in WHO grade I tumors, median 10.92% in GII meningiomas, and 4.99% in GIII meningiomas, respectively, as presented in Figure 3C.

## 4. Discussion

Whole-transcriptome gene expression analysis with microarrays or RNA-seq applied in multiple previous studies focused on meningiomas [20,21,22,23,24,25,26,27,28,29,30,31,32,33,34,35]. Aberrant gene expression was investigated in diverse contexts, including the search for clinically relevant markers [20,21,22,23], exploring tumor biology [36,37,38], and as part of developing multi-omic classification systems [5,6]. For years, the WHO grade has remained a fundamental feature of the histological classification of meningiomas, alongside the recent introduction of new evidence-based molecular markers. Given that a higher grade corresponds to progressively more aggressive tumor growth, the objective of this work was to examine gene expression relative to tumor aggressiveness in the most common histological subtypes. We intentionally excluded WHO GI transitional meningiomas, as these tumors are, by definition, mixed and composed of two other common GI subtypes, namely meningothelial and fibrous meningiomas [2]. Unsupervised transcriptome analysis revealed three clusters that largely corresponded to histological subtypes including clusters composed mostly of meningothelial WHO GI (cluster M), fibrous GI (cluster F), and cluster containing most of atypical GII and anaplastic GIII tumors (cluster A). Thus, atypical and anaplastic tumors appeared more similar to each other than two subtypes of benign meningiomas. Similar clustering was observed in the analysis of combined publicly available RNA-seq datasets including a large group of meningiomas, where most of WHO GII and GIII tumors concentrated within a single cluster, while the majority of benign tumors formed additional clusters [39]. However, such clustering was not observed in another transcriptomic study [40]. The transcriptome-based distinction between meningothelial and fibrous meningiomas reflects well-described differences in the molecular profiles of these tumors. It is well established that fibrous meningiomas commonly harbor *NF2* mutations or chromosomal loss, whereas meningothelial tumors frequently carry *TRAF7* and *AKT1* mutations [41]. Separate clustering of *NF2*- and *TRAF7*-mutated meningiomas has recently been documented in a large patient cohort [39]. These two subtypes of benign meningiomas also form distinct clusters in analyses of whole-genome methylation data [42]. In order to investigate grade-related differences in gene expression, we used tumor WHO grading as numeric values to identify genes showing a significant progressive increase or decrease in expression from benign to anaplastic tumors. To interpret the role of the large number of DEGs identified in this comparison, we applied Gene Set Enrichment Analysis (GSEA). As expected, higher-grade meningiomas showed upregulation of processes related to cell proliferation, consistent with basic histological markers of atypical and anaplastic meningiomas [2]. Another category of processes highly upregulated in higher-grade tumors was related to cellular metabolism. Metabolic reprogramming is recognized as a hallmark of cancer [43,44], particularly in aerobic glycolysis, known as the Warburg effect [44] and it also plays a role in aggressive meningioma growth [45,46,47,48]. Our analysis identified *PGK1* expression as the most significantly correlated with meningioma grade. *PGK1* encodes a phosphoglycerate kinase, a key enzyme in glycolysis, including the anaerobic glycolysis pathway [49]. *PGK1* drives tumorigenesis primarily through glucose metabolism, but it also contributes to a variety of processes such as angiogenesis, epithelial–mesenchymal transition, mitochondrial metabolism, and DNA repair and replication [49]. *PGK1* is included in the recently developed 34-gene expression signature for predicting meningioma recurrence and therapy response [23,50]. When predicting the recurrence in WHO grade I meningiomas, PGK1 expression shows the largest effect size [51]. Given that *PGK1* was identified as one of the top differentially expressed genes (DEGs) in our study, we further assessed phosphoglycerate kinase expression, which, to our knowledge, has not been previously documented in the literature. Phosphoglycerate kinase 1, a housekeeping gene, was detected in all samples; however, its protein level was markedly elevated in atypical and anaplastic meningiomas compared to benign tumors, reinforcing the notion of increased metabolic adaptation in high-grade meningiomas.

High-grade meningiomas were characterized by the downregulation of processes related to cell morphogenesis and immune regulation, suggesting a dedifferentiated tumor cell phenotype and reprogramming of the tumor immune microenvironment. Interestingly, we also observed significant downregulation of ion transport-related pathways, with genes encoding potassium and calcium channels (*KCNMA1*, *CACNA1D*, and *CACNA2D1*) among the top DEGs. Lower *KCNMA1* expression in high-grade compared to benign meningiomas has been previously reported in oligonucleotide microarray-based studies [24,52,53] and this gene was also included in a 49-gene prognostic expression signature [30]. Meningioma cells are recognized as electrically non-excitable [54]; therefore, in these tumors, the role of ion channels is rather associated with the regulation of cellular signaling. Intracellular calcium is one of the key (?) second messengers, and its levels are influenced by membrane potassium and calcium channels. Ca^2+^ signaling has been shown to promote growth and tumorigenesis in brain cancers and meningiomas [55]. In meningioma cells, calcium signaling was also recently found to contribute to apoptosis induced by NF2, a well-known tumor suppressor in meningiomas [56].

RNA-seq results can provide insight into the tissue cellular heterogeneity [18]. Using deconvolution of bulk RNA-seq data informed by scRNA-seq profiles, we identified grade-dependent differences in tumor microenvironment composition. Most importantly, our results show a gradual depletion of border-associated macrophages (BAMs) across WHO grades, with the lowest abundance observed in anaplastic meningiomas. BAMs, together with microglia, constitute a class of resident macrophages within brain tissue [57]. While microglia primarily reside in the brain parenchyma, BAMs are predominantly located at the blood–brain barrier, the blood–cerebrospinal fluid barrier, and within the meninges [58]. Unlike microglia, BAMs do not appear to be directly involved in glioma pathogenesis, as their abundance remains similar in naïve and glioma-bearing mouse brains [59]. However, they are thought to facilitate colonization of the dura and leptomeninges by brain metastases [57]. The lower content of BAMs in higher-grade meningiomas corresponds to downregulation of immune signaling process revealed by GSEA.

The role of BAMs in meningioma remains unclear. Notably, our deconvolution analysis provides preliminary evidence of a reduction in these macrophages in higher-grade tumors. Further research is needed to clarify the role of this macrophage population, considering the reported complexity of meningioma-associated macrophage subsets [60,61].

We observed increased granulocyte abundance in atypical and anaplastic meningiomas compared with benign tumors. This observation may indicate a shift toward a protumor inflammatory and immunosuppressive microenvironment, as granulocytes can promote angiogenesis, matrix remodeling, and immune evasion in cancer [62]. Alternatively, their presence may reflect greater necrosis or hypoxia in higher-grade tumors [62]. In recent years, the heterogeneity of cancer- associated granulocytes has been recognized, with these cells exhibiting both pro- and anti-tumorigenic activities [63].

Several limitations of this study should be noted, including the relatively small sample size and the subjective, retrospective selection of patients based on histological diagnosis. Although this selection was justified by the study’s objectives, it limits the generalizability of the findings to the general patient population. The results of the deconvolution analysis used to describe the tumor microenvironment should be considered preliminary. Overall, our findings are observational rather than experimental. While we describe tumor grade-related biological features from a transcriptomic perspective, causality cannot be inferred. Further functional and experimental studies and investigations are needed to clarify the roles of specific genes, pathways, and microenvironmental components in the mechanisms underlying the acquisition of aggressive growth potential.

## 5. Conclusions

In this study, we used RNA-seq to determine gene expression patterns in benign (meningothelial and fibrous), atypical, and anaplastic meningiomas and to identify genes with grade-related expression. We identified key genes and pathways dysregulated in high-grade meningiomas, characterized by enhanced cell proliferation and metabolism, disrupted ion transport, and altered immune regulation. The analysis of intratumoral heterogeneity using RNA-seq deconvolution indicated a potential role for the reduction in border-associated macrophage numbers in the development of an aggressive tumor phenotype.

## Figures and Tables

**Figure 1 cancers-17-03324-f001:**
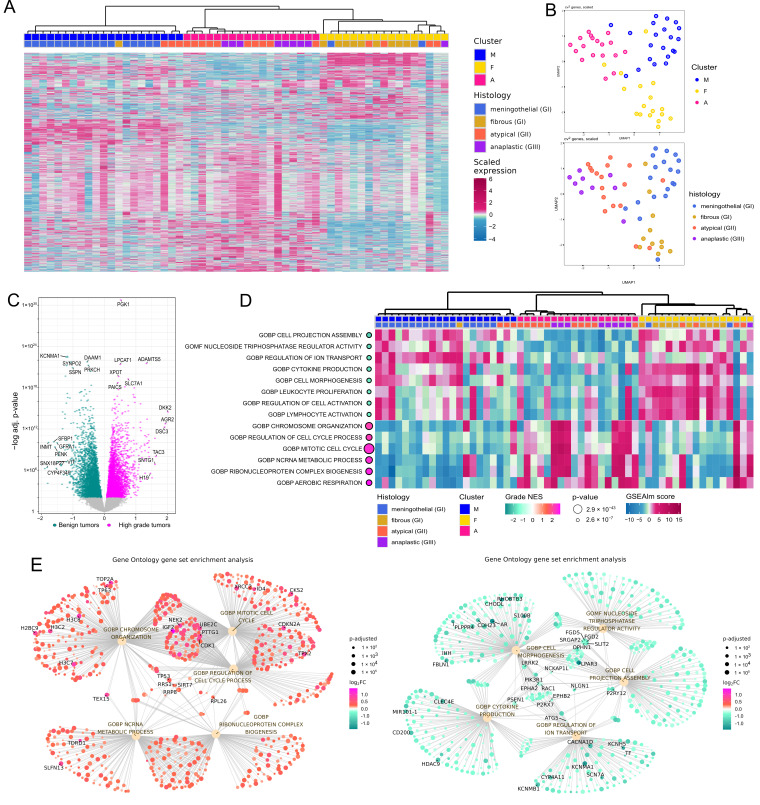
Results of RNA sequencing and gene expression analysis in meningiomas. (**A**) Clustering the meningiomas with three clusters largely reflecting the histological subtypes. (**B**) Uniform manifold approximation and projection (UMAP) analysis of meningioma samples. (**C**) Volcano plot showing genes differentially expressed with respect to WHO grade. (**D**) Gene ontology-based gene set enrichment (GSEA) of identified DEGs, with a heatmap representing the expression of genes annotated to specific pathways in each tumor sample. (**E**) Gene networks (?) illustrating relationships between activities of genes of top processes upregulated (in red) and downregulated (in green) in higher-grade meningiomas.

**Figure 2 cancers-17-03324-f002:**
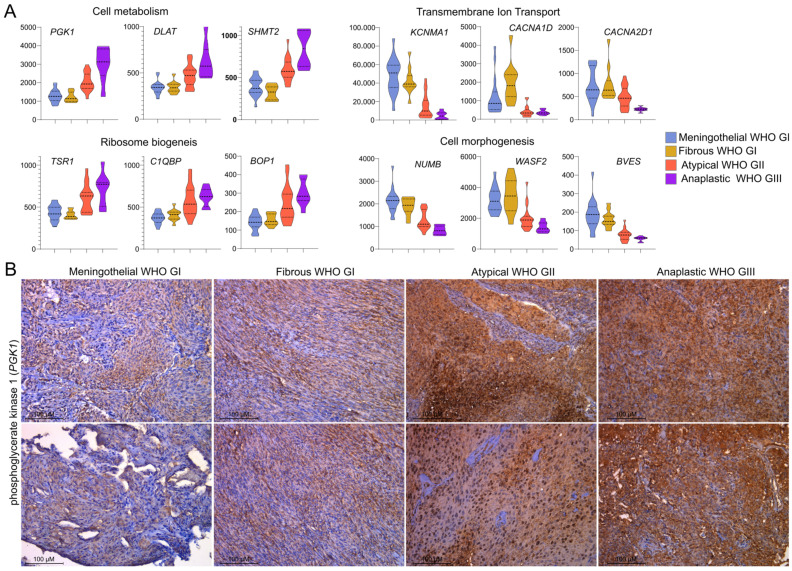
Expression of selected genes with grade-related expression patterns. (**A**) Expression levels of DEGs annotated to selected gene ontology processes. (**B**) Representative examples of immunohistochemical staining of phosphoglycerate kinase 1 (PGK1), showing increased expression in atypical (WHO grade II) and anaplastic (WHO grade III) meningiomas.

**Figure 3 cancers-17-03324-f003:**
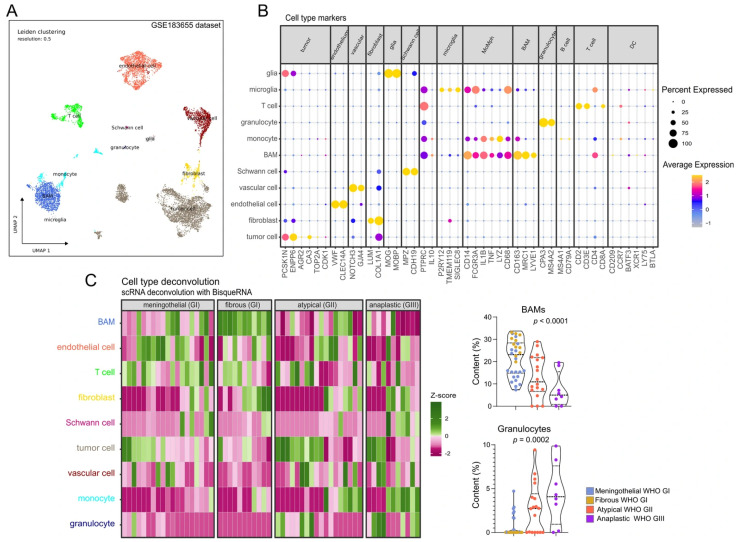
Analysis of WHO grade-related intratumor heterogeneity in meningiomas. (**A**) Reanalysis of the publicly available scRNA-seq dataset (GSE183655) to determine the cellular composition of meningioma. (**B**) Expression levels of marker genes in each identified cell type in meningioma. (**C**) RNA-seq deconvolution-based assessment of the content of each cell type in meningioma samples analyzed by RNA-seq.

**Table 1 cancers-17-03324-t001:** Characteristics of the patients.

Patients Clinical/Demographical Feature	
Number of patients	60
Females	38/60 (63.33%)
Males	22/62 (36.67%)
Age at surgery (years; median (range))	62.5 (38–88)
Meningioma histological subtype	
WHO grade I	30/60 (50%)
Meningothelial	19/60 (31.67%)
Fibrous	11/60 (18.33%)
WHO grade II, Atypical	18/60 (30%)
WHO grade III, Anaplastic	12/60 (20%)

**Table 2 cancers-17-03324-t002:** Results of immunohistochemical staining of meningiomas for phospho-glycerate kinase 1 (PGK1) expression.

Histological Subtype	No Expression (No. of Samples)	Weak Expression (No. of Samples)	Moderate Expression (No. of Samples)	High Expression (No. of Samples)
WHO grade I Meningothelial (*n* = 6)	0	4	1	1
WHO grade I Fibrous (*n* = 6)	0	4	2	0
WHO grade II, Atypical (*n* = 6)	0	0	3	3
WHO grade III, Anaplastic (*n* = 6)	0	0	2	4

## Data Availability

RNA-seq data on meningioma FFPE samples are available at ZENODO (DOI: 10.5281/zenodo.10808492).

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
