# Peer review of "Determining the Biological Features of Aggressive Meningioma Growth with Transcriptomic Profiling"

_cancers, 2025, doi:10.3390/cancers17203324_

Round 1

Reviewer 1 Report

Comments and Suggestions for Authors

The artilce submitted to Cancers j entitled “Determining the biological features of aggressive meningioma growth with transcriptomic profiling” by Baluszek et al., 2025.

The authors identified genes with grade-related expression and subsequently interpreted their roles through analysis of biological pathways and estimated the cellular composition of the heterogeneous tumor samples. The results reflect well-known features of aggressive growth, such as increased cell proliferation, but also highlight less- characterized aspects of tumor biology. These include alterations in cellular metabolism, with a role for phosphoglycerate kinase 1, changes in transmembrane ion transport, and differences in the content of border-associated macrophages, which represent key distinctions between benign and aggressive meningiomas.

  1. Introduction:

The passage gives a general clinical and pathological overview of meningiomas, summarizing prevalence, grading, histology, prognosis, and associated biology. While informative, there are several scientific, logical, and writing-related faults that reduce the clarity, accuracy, and scientific rigor of the text.

What is “Benign Malignancies”?

WHO classification uses “Grade I” or “WHO Grade I,” not “(G) I.”

Fix the statement: “In rare cases, benign meningiomas can recur with more aggressive histological features, particularly after incomplete resection or radiation.”

Mention that molecular alterations are now incorporated into the grading of meningiomas, reflecting prognosis beyond histology alone.

“Basically associated” is imprecise and redundant; the sentence could be more concise, per this reduces readability and scientific tone.

Specify: “These histological features reflect underlying molecular changes, such as chromosomal instability, altered cell cycle regulation, and epigenetic dysregulation.” no need for filler sentences.

Repetition of References shouldn’t be there.

The passage ends with a general statement about "molecular abnormalities" without transitioning to how that motivates molecular or transcriptomic studies. End with “As such, understanding the transcriptomic and molecular landscape of meningiomas is critical to uncovering the mechanisms underlying tumor aggressiveness and recurrence.”

  1. Aim:

Typographical or Terminological Errors

Example: "GIIII" should be GIII (WHO Grade III)

The authors wrote "Scientific efforts have been focused on characterization of molecular changes... for description of possible pathogenic mechanisms and... markers..." The statement is broad and lacks specificity about what has already been well-characterized and what knowledge gaps remain.

The authors claim "...identifying the role of genetic, epigenetic, transcriptomic and proteomic changes..." But, the sentence lumps multiple omics levels together without specifying which aspects (e.g., specific genes, pathways, or molecular alterations) have been previously identified.

No Mention of Validation or Functional Follow-up,

The scope is broad and risks being descriptive rather than hypothesis-driven.

  1. Methodology:

Limited sample size for each tumor subtype that should be acknowledged in a limitation part, that limits the conclusion obtained regarding generalization of the results

Histological tumor calssification seems to be subjective, if not please clarify,

Differentianl results are not clear,

Bulk RNA-seq was used, with subsequent deconvolution using scRNA-seq data.

Please, emphsize on cell-type contribution, and tumor stages,

The clustering results are data-driven artifacts rather than biologically meaningful separations (this to be added to limitations),

Focusing heavily on PGK1 which is a single gene overlook the polygenic nature of tumor grade progression.

PGK1 is a common "housekeeping"  glycolytic enzyme often upregulated in many cancers,

Without the same patients samples as tumors progress, it's hard to infer causality or progression pathways.

  1. Results:

No link to survival results, therefore, it's unclear how functionally relevant or prognostic the identified gene signatures are?

There's no mention of batch effect correction or normalization procedures for RNA-seq.

  1. Discussion:

Needs more elaboration,

No Mention of Clinical Relevance

Overinterpretation of Correlation as Causation

Tumor Subtypes Grouped Without Justification

Author Response

The replies to the Reviewer's comment were uploaded in separate file

Reviewer 2 Report

Comments and Suggestions for Authors

 Baluszeket al performed transcriptomic profiling of various grades of meningioma samples using RNAseq. They used 60 samples of different grades and compared them to identify genes that are specific to each grade. They compared these differentially expressed genes with deposited scRNAseq data. After rigorous analysis they assigned gene functions related to aggressive grade and identified PGK1 as a significant grade associated gene in meningioma.

This study is a comprehensive transcriptomic analysis of meningioma, and the experiments and analysis are performed well. However, presentations of results are poor and there are certain concerns that need to be addressed.

  1. In materials and methods, immunohistochemistry: More detailed information (microscope information, number of each observation, statistical considerations etc)  about visualization of stained slides are necessary.
  2. Authors should mention all information about earlier all transcription prefilling using meningioma samples with references.
  3. The representation in Figure 1 is very poor-resolution, texts within the figures and too much information in 1 figure. Authors should split the figure1 into at least two figures to make larger image of each and texts to higher resolution.
  4. In figure1,GSEA analysis, the distinct use of Lipid localization, alcohol metabolic and biosynthetic pathway from GRAD1 to aggressive form should need special attention and should be highlighted discussion.
  5. Figure 2, the texts within the figure should be improved to larger font and higher resolution.
  6. In figure 2, significant decrease of monocyte (may be fibroblast also) population in anaplastic group should also be discussed.

Author Response

The replies to the Reviewer's comment were uploaded in separate file.

Round 2

Reviewer 1 Report

Comments and Suggestions for Authors

The manuscript is now improved

Reviewer 2 Report

Comments and Suggestions for Authors

The quality of the paper is improved.
